# Ship Detection for Optical Remote Sensing Images Based on Visual Attention Enhanced Network

**DOI:** 10.3390/s19102271

**Published:** 2019-05-16

**Authors:** Fukun Bi, Jinyuan Hou, Liang Chen, Zhihua Yang, Yanping Wang

**Affiliations:** 1School of Information Science and Technology, North China University of Technology, Beijing 100144, China; bifukun@163.com (F.B.); jinyuan862@163.com (J.H.); yangzhtyy@163.com (Z.Y.); 2School of Information and Electronics, Beijing Institute of Technology, Beijing 100081, China; cl_bl2010@sina.com

**Keywords:** scene classification, ship detection, visual attention enhanced network, DSOD

## Abstract

Ship detection plays a significant role in military and civil fields. Although some state-of-the-art detection methods, based on convolutional neural networks (CNN) have certain advantages, they still cannot solve the challenge well, including the large size of images, complex scene structure, a large amount of false alarm interference, and inshore ships. This paper proposes a ship detection method from optical remote sensing images, based on visual attention enhanced network. To effectively reduce false alarm in non-ship area and improve the detection efficiency from remote sensing images, we developed a light-weight local candidate scene network($L^{2}$CSN) to extract the local candidate scenes with ships. Then, for the selected local candidate scenes, we propose a ship detection method, based on the visual attention DSOD(VA-DSOD). Here, to enhance the detection performance and positioning accuracy of inshore ships, we both extract semantic features, based on DSOD and embed a visual attention enhanced network in DSOD to extract the visual features. We test the detection method on a large number of typical remote sensing datasets, which consist of Google Earth images and GaoFen-2 images. We regard the state-of-the-art method [sliding window DSOD (SW+DSOD)] as a baseline, which achieves the average precision (AP) of 82.33%. The AP of the proposed method increases by 7.53%. The detection and location performance of our proposed method outperforms the baseline in complex remote sensing scenes.

## 1. Introduction

With the rapid development of remote sensing technology, ship detection plays a significant role in both military and civil fields, such as military port investigation, dynamic port monitoring, fishery management, and maritime rescue [1,2,3,4,5]. However, ship detection from remote sensing images often have the large size of image, and the applications for ship detection require high interpretation timeliness. In addition, the detection from remote sensing images has some problems, including complex scene structures, a large amount of false alarm interference and difficulties detecting inshore ships [6,7]. Meanwhile, ships in remote sensing scenes have scale difference between ships and arbitrary-orientation, which make them difficult to accurately detect and locate.

Researchers in this field have proposed a series of methods. Traditional detection methods [8,9,10,11,12]: Chao Dong et al. [11] constructed a novel visual saliency detection method to locate candidate regions, and a trainable Gaussian support vector machine (SVM) classifier was performed to validate real ships out of ship candidates. Fang Xu et al. [12] proposed a novel ship detection method from coarse to fine, which implemented a global saliency model and multi-level discrimination method to detect ships. Previously, the author of this paper used a bottom-up visual attention mechanism to select salient candidate regions across entire detection scene, and two complementary types of top-down cues were employed in order to discriminate among the selected ship candidates [13]. In addition, it also used an omnidirectional intersected two-dimensional scanning strategy to rapidly extract candidate regions, and a decision mixture model was proposed to identify real ships from candidate objects [14]. The above traditional detection methods mainly have two steps. (1) Extracting the ship candidate area, because the scene of ship detection is complex and false alarms tend to occur in areas where ships are not present, which may result in the ship being missed. (2) In the ship identification stage from the candidate region, remote sensing images have a large amount of interferences with similar characteristics in colors, shapes, and textures, such as docks, warehouses, and some strip buildings. Traditional detection methods have limited feature extraction ability, docked ship and side-by-side ships are liable to be missed. Ships that are berthed at dock, which is called “docked ship”. Ships docked side by side, which is called “side-by-side ship”.

In recent years, the convolutional neural network (CNN) has been widely used for feature extraction. To better characterize the objects’ features, many excellent deep learning network structures have strong robustness, such as Faster-RCNN [15], You Only Look Once (YOLO) [16], and Single Shot MultiBox Detector (SSD) [17]. Currently, the CNN can superimpose the network depth to improve the feature representation. For example, the Resnet architecture increases the width of the network to generalize the network [18], and ResnetXt increases the cardinality of the network to get deeper level semantic features [19]. The CNN also has some drawbacks, training the model parameters of the CNN requires a large number of training samples. Where the number of training samples are limited, such as linear discriminant analysis [20], support vector machine [21], principal coefficients embedding (PCE) [22], and a method of inshore ship detection using DPM (once proposed by the authors of this paper) [14], which have less parameters to train, and trained the results with a small sample may outperform CNN. However, compared with the ship classification task, the samples of ship detection tasks are relatively sufficient. After testing, in some complex remote sensing scenes, the CNN can achieve better detection results than typical methods. In traditional CNN, the convolutional layers are used to extract the objects’ features, and the fully connected layers are used to predict the labels of the object. The methods-based CNN have the ability to obtain high-level features, and the CNN has recently been introduced into various tasks of remote sensing, such as object detection [23,24,25,26], object recognition [27,28], and scene analysis [29,30]. In the field of ship detection, using optical remote sensing images, Zenghui Zhang et al. [31] proposed rotated region proposal networks to generate multiple orientated proposals, with ship orientation angle information, and regressed the orientation angles of the bounding box. Wenchao Liu et al. [32] designed a network with feature maps that used the layers with different depths, and the used orientation angle information to predict the bounding box. First, the ship is often detected in a special scene, such as in the river, port, sea, etc. The ship detection methods, with CNN, generally detect ships from the whole remote sensing image [33,34]. Scene extraction before detection can improve detection efficiency and reduce false alarms in scenes without a ship. Then, inshore ship detection is still a challenging task in ship detection. The shallow CNN can only extract simple features, and superimposing the depth of the convolutional network can generally enhance the feature differences between a ship and its background. Ship detection from remote sensing images often have a large size of an image, whereas large amount of convolution calculation reduce the efficiency of ship detection. So a method of inshore ship detection is to embed an enhanced features module in CNN, which does not add a lot of computation and can enhance the feature differences between a ship and its background. In object detection with traditional CNN, horizontal bounding box regression is often used to predict the position of the object [24,35]. Ships in remote sensing scenes have characteristic of long strip shape, arbitrary-orientation, and being densely docked at the port. The horizontal bounding box is not suitable for accurately locating the position of ship.

This paper proposes a ship detection method from optical remote sensing images based on the visual attention enhanced network. The main contributions of this paper mainly lie in the following two aspects.

(1)We develop a lightweight local candidate scene network($L^{2}$CSN), which extracts local candidate scenes with ship and eliminates some interference scene areas without ship. The $L^{2}$CSN reduces the false alarms in non-ship areas and improves the detection efficiency from remote sensing images.(2)In the stage of ship detection from the local candidate scene, we propose a ship detection method based on the visual attention DSOD(VA-DSOD). To extract the deep level features, we both use DSOD to extract the semantic features, and embed the visual attention enhanced network in DSOD to extract visual features, which improves detection performance, especially for docked ships and side-by-side ships. In addition, the rotated bounding box regression, based on VA-DSOD, enhances the positioning accuracy.

This paper is organized as follows. In Section 2, we introduce the details of the proposed method. In Section 3, we conduct qualitative and quantitative experiments to evaluate the performance of the proposed method. Section 4 concludes this paper.

## 2. Proposed Method

In this section, we describe the algorithm’s framework of the proposed method and its key modules. As shown in Figure 1, the algorithm’s framework has two parts: Local candidate scenes extraction, based on $L^{2}$CSN and ship detection from the local candidate scenes based on VA-DSOD. The first part adapted the $L^{2}$CSN to extract local candidate scenes from remote sensing image. The second part accurately predicted the ships in the selected local candidate scenes, and the detection results in the local candidate scene are mapped back to the original image to locate the positions of ships.

### 2.1. $L^{2}$CSN Architecture

Remote sensing images have complex scenes and ships are generally sparsely distributed in the images. Efficient local candidate scene extraction is essential for detecting ships from remote sensing images, so we designed the $L^{2}$CSN to extract local candidate scenes. In this paper, motivated by Inception-v4 [36], $L^{2}$CSN adopted the stem network structure. This stem can effectively improve the feature expression ability, without adding computational cost too much, which is better than other more expensive methods. According to the characteristics of ships in the scene, ships generally exist at sea, on rivers and along coast. Therefore, we divided the training samples of the $L^{2}$CSN into four categories: Seas, rivers, coasts, and land.

The $L^{2}$CSN architecture is shown in Figure 2. In the remote sensing images, the sizes ranged from 5000 × 5000 pixels to 15000 × 15000 pixels, and were divided into local scene areas (with 1000 × 1000 pixels) with partial overlaps. The overlap was to prevent cutoff ships from being missed. Since the ship pixels in the remote sensing images are relatively small, the overlap area can be adjusted according to the maximum size of ships. The local candidate scene extraction used the structural information of a scene, and can be sampled and then classified. Therefore, before each local scene area is input into the $L^{2}$CSN, they are sampled from 1000 × 1000 pixels to 112 × 112 pixels, thereby reducing the calculation. Net1 and Net2 in the $L^{2}$CSN have similar structures, which guaranteed the effectiveness of the $L^{2}$CSN by reducing the convolution computations and adjusting the network structure. In Net1, a network branch continued to convolve to extract features. The other branch conducted the pooling for the feature maps and made the cross layer connection with subsequent feature maps, which combined the information from feature maps at different scales. We used the 1 × 1 convolution kernel to conduct the dimensionality reduction processing and ReLU and batch normalization to enhance the non-linearization of the features. The last layer of the $L^{2}$CSN used the fully connected layer and the soft-max function to generate the classification score of local scenes, and we took the maximum score as the discriminant criterion of the extracted the local scene. According to the properties of the local scene, we defined the local candidate scene, which is classified as coast, river, and sea. Land was defined as the local non-candidate scene.

### 2.2. Ship Detection from Local Candidate Scenes Based on VA-DSOD 

The flow chart of the ship detection, based on VA-DSOD, is shown in Figure 3. The detection network has two parts. (1) For the extracted local candidate scenes, we extracted the deep semantic features and visual features based on VA-DSOD. (2) We predicted the rotated bounding box of ships in the feature maps of different scales.

#### 2.2.1. Semantic and Visual Features Extraction Based on VA-DSOD

Compared with the application of sufficient samples, such as face detection, the number of ship samples in remote sensing field are relatively limited. However, the pre-training phase of many typical network frameworks required a large number of training samples. In this paper, the DSOD structure is used as the feature extraction module [37], since its network structure can train and fit the best model without the base network of pre-training. The densenet in DSOD extract semantic features and we reduced part of the redundant convolution structure according to the structural characteristics of ships. Although the features that are extracted by DSOD are very rich, the features differences between the ships and some interference background are not very significant, such as docked ships and side-by-side ships in inshore, which cause the ships to be missed in the local candidate scenes. To further improve the detection accuracy and reduce the positioning deviations of the rotated bounding box, we embeded the visual attention enhanced network in DSOD to structure the visual attention DSOD(VA-DSOD). In VA-DSOD, the semantic and visual features can be extracted.

The visual attention enhanced network is shown in Figure 3. The input feature map F adopts the visual attention enhanced network to enhance the visual attention in the channel dimension and enhance the saliency of ship for local features. The embedding method in the DSOD is as follows. To obtain the channel attention map, we squeezed the spatial dimension of the input feature map by using the average pooling and the maximum pooling, thereby generating two different spatial context descriptors: $F_{avg}^{c}$ and $F_{max}^{c}$, which represent average-pooled features and max-pooled features in the convolutional layers. Then, these features are converted to a shared convolution layer. The shared network is constituted of multi-layer perceptron (*MLP)*. Attention maps $M_{1}$ and $M_{2}$ are merged into a one-dimensional channel attention map $M_{c}$ with feature maps information. Finally, $M_{c}$ is multiplied by the input feature map F to form a feature map with stronger representational ability [38].
(1)$$M_{c}\left(F\right)=\sigma \left(MLP\left(AvgPool\left(F\right)\right)+MLP\left(MaxPool\left(F\right)\right)\right)\;=\sigma \left(W_{1}\left(W_{0}\left(F_{avg}^{c}\right)\right)+W_{1}\left(W_{0}\left(F_{max}^{c}\right))\right)\right)$$
where σ denotes the sigmoid function, *MLP* is the weights, $M_{0}$ and $M_{1}$ are shared for both inputs and the ReLU activation function is followed by $M_{0}$.

#### 2.2.2. Rotated Bounding Box Regression

Ships in remote sensing scenes have scale difference and arbitrary-orientation. In the DSOD architecture, the multi-scale pyramid solves the problem of scale difference, but the horizontal bounding box cannot accurately locate the strip and arbitrary-oriented ships, and side-by-side ships are also easily combined. Therefore, to predict arbitrary-oriented ships, the rotated bounding box regression is carried out on feature layers of different scales. In this paper, the visual attention enhanced network highlights the local features of ships to reduce the positioning deviation, and the detected ships’ coordinates, are mapped back to the original image. However, the overlapping area is generated when the local candidate scene is extracted, and the ships within it are repeatedly detected. In addition, each pixel of the feature maps has multiple default box, and the same position can predict multiple detection results. We used the rotated non-maximum suppression(NMS) to eliminate the overlapping detection results.

In the DSOD [37], to adapt to various objects in the natural scene, the object is predicted on the six scales feature map and default boxes set as the small aspect ratios. The aspect ratios of the default boxes are (1:1, 1:1, 1:2, 2:1, 1:3, 3:1, 5:1, 1:5) at each location. We set the five scales feature maps according to the scale characteristics of the ships, and the size of the scale is {64 × 64, 32 × 32, 16 × 16, 8 × 8, 4 × 4} pixels on {C1, C2, C3, C4, C5}. In addition, different from general objects, the ship are generally a long strip and tend to have large aspect ratios. To further accommodate ships with different aspect ratios, we set 8 default boxes on each pixel of the feature map according to the ship’s proportional characteristics. The aspect ratios of the default boxes were set as (1:1, 1:1, 1:3, 3:1, 1:5, 5:1, 1:7, 7:1), which cover the different proportions of ships.

In the training phase, according to the box overlap following the matching scheme in [17], the ground truth box was matched to the correct default box $d_{0}\;$= ($x_{0},y_{0},\;w_{0},h_{0}$) at each location. $x_{0}\;\mathrm{and}\;y_{0}$ are the center point coordinates; $w_{0}\;\mathrm{and}\;h_{0}$ are the width and height. The four-point coordinates of default box, d = ($x_{1}^{d},y_{1}^{d},x_{2}^{d},y_{2}^{d},x_{3}^{d},y_{3}^{d},x_{4}^{d},y_{4}^{d}$), are obtained by (2) [39]. The network uses the gradient descent algorithm to obtain the error between the default box and ground truth box, and continuously learns to obtain the final convolution kernel parameters. In the detection phase, as shown in Figure 4, to predict the rotated bounding box and classification score of ships, the trained convolution kernel parameters were subjected to a series of convolutions on the image. The rotated bounding box is achieved by predicting the regression of offsets Δq from a number of default boxes at each location.  Δq=($\Delta x_{1}$,$\;\Delta y_{1},\Delta x_{2},\;\Delta y_{2},\Delta x_{3},\;\Delta y_{3},\Delta x_{4},\;\Delta y_{4}$). The final coordinate of the rotated bounding box are calculated as q = ($x_{1}^{q},y_{1}^{q},x_{2}^{q},y_{2}^{q},x_{3}^{q},y_{3}^{q},x_{4}^{q},y_{4}^{q}$) by (3).

(2)$$x_{1}^{d}=x_{0}-w_{0}/2,y_{1}^{d}=x_{0}-h_{0}/2\;x_{2}^{d}=x_{0}+w_{0}/2,y_{2}^{d}=y_{0}-h_{0}/2$$

(3)$$x_{3}^{d}=x_{0}+w_{0}/2,y_{3}^{d}=y_{0}+h_{0}/2\;x_{4}^{d}=x_{0}-w_{0}/2,y_{4}^{d}=y_{0}+h_{0}/2\;x_{n}^{q}=x_{n}^{d}+w_{0}\Delta x_{n},n=1,2,3,4\;y_{n}^{q}=y_{n}^{d}+h_{0}\Delta y_{n},\;n=1,2,3,4$$

The rotated NMS determines whether the two rotated bounding boxes are overlapped, and if they are overlapped, we calculate their Intersection over Union (IoU). As shown in Figure 5, the overlapped area can be computed using triangulation, and we can obtain the overlapped area So and the union area Su as follows:(4)$$S_{o}=S_{\Delta P_{1}P_{2}P_{3}}+S_{\Delta P_{1}P_{3}P_{4}}+S_{\Delta P_{1}P_{4}P_{5}}$$
(5)$$S_{u}=w_{0}*h_{0}+w_{1}*h_{1}-S_{o}$$

The skew IoU as follows:(6)$$IoU=S_{o}/S_{u}$$

## 3. Experiments and Discussion

In this section, using a large number of typical remote sensing dataset with ship scenes, we conduct qualitative and quantitative experiments to evaluate the performance of the proposed method. For extracting the results of local candidate scenes, we manage to demonstrate the adaptability of the $L^{2}$CSN in complex scenes. To illustrate the effectiveness of the features, we compare the detection accuracy and positioning accuracy between VA-DSOD and DSOD. By comparing it with other state-of-the-art methods, we analyze the detection performance and timeliness of the proposed method. Our proposed method is implemented on Caffe, trained and tested on the Ubuntu16.04 operating system with a GTX Titan X GPU (12 GB on board memory).

### 3.1. Experimental Dataset

Our experimental dataset was collected from publicly available Google Earth and GaoFen-2 satellite with 6410 optical images, a resolution of 1 m and the images size range from 5000 × 5000 pixels to 15000 × 15000 pixels, which are downloaded using the software of Notation on Waterways. Among these, 4520 images are obtained from Google Earth, the other 1890 images are obtained from GaoFen-2, and images contain a variety of scenes and different types of ships. For the training set of the $L^{2}$CSN, we randomly select 5000 images from all experimental datasets, which are cut into 1000 × 1000 pixels and manually divided into four categories: land, coast, river and sea. The remaining 1410 images are used as test set. For the dataset of ship detection based on VA-DSOD, we randomly select 5000 images as the training set. To match the input size of the local candidate scenes, we cut the training set images 1000 × 1000 pixels, which are marked as rotated bounding box. The label is (*x1, y1, x2, y2, x3, y3, x4, y4, c*); x and y are the horizontal and vertical coordinates, respectively; and c is the class. Since the scales characteristics of ships vary greatly in remote sensing images, the division criteria of *c* is the scale difference of ships. Due to the large number of training images required, which are manually classified and marked by the data marking company (BasicFinder Ltd, Beijing, China).

### 3.2. Experiment Results and Analysis for the $L^{2}$CSN

To assess the effectiveness and reliability of the $L^{2}$CSN, we conduct qualitative and quantitative experiments and analysis for the extracted local candidate scenes.

(1) Qualitative Experiment and Analysis: 

We extracted local candidate scenes from a remote sensing image (6370 × 11860 pixels) of typical scene and map the classification results back to the image, as shown in Figure 6. We adopted the traversal strategy of the sliding window, and the experiment results show that the overlapping area in A overcame the missed ships using the sliding window. The local candidate scenes with a few port area are probably misclassified as local non-candidate scenes, the overlapping part B overcame the missed ships from misclassified local candidate scenes. The $L^{2}$CSN effectively extracts the local candidate scenes and largely eliminates the local non-candidate scenes for a variety of complex scenes.

(2) Quantitative Experiment and Analysis: 

Since ship detection relies on local candidate scenes, the local candidate scene to be misclassified directly causes the ship to be missed. The confidence represents the classification score to classify the scene, which proves the credibility of the scene being correctly classified and, the change confidence can affect the detection performance and timeliness. To reliably extract the local candidate scenes, we perform a *recall rate*
$\left(R_{s}\right)$ analysis of the local candidate scenes in the test set and determined the confidence of the $L^{2}$CSN. The *recall rate* ($R_{s}$) is calculated as follow.
(7)$$R_{s}=\frac{TP^{s}}{TP^{s}+FN^{s}}=\frac{\sum_{i}^{N}TP_{i}^{s}}{\sum_{i}^{N}(TP_{i}^{s}+FN_{i}^{s})}$$
where $TP^{s}$ represents the number of correctly classed local candidate scenes, $FN^{s}$ represents the number of missed local candidate scenes, and *N* indicates the number of local candidate scenes. The $R_{s}$ curves of the local candidate scene is shown in Figure 7. When the confidence is greater than 0.3, the $R_{s}$ will gradually decrease. The number of missed ships in local candidate scenes will increase sharply, which could lead to ships to be missed. If the confidence is less than 0.3, it may increase the number of local non-candidate scenes, which will generate many false alarm and redundant computations. Because the threshold of 0.3 is an inflection point, we select 0.3 as the final threshold of the $L^{2}$CSN to balance performance and efficiency.

### 3.3. Experimental Results and Analysis for the VA-DSOD

To demonstrate the effect of visual attention enhanced network on ship detection, we conducted training and testing for DSOD, and VA-DSOD in typical scenes, respectively. The typical test results are shown in Figure 8. Although DSOD utilizes feature information, the detection result is still poor in the complex scenes. These scenes often contained some false alarms with the similar characteristic to ship (such as dock, small island, etc.), which tend to be misidentified as ship in Figure 8(a1). For the side-by-side ships and docked ships, the feature difference between the ship and its background is weak, several ships are even missed in Figure 8(b1), and the location of ships have a certain deviation in Figure 8(c1). Compared with the DSOD, the experimental results of VA-DSOD indicate that the performance of detection and location have been improved in complex scenes. As shown in Figure 8(a2,b2), the false alarms can be largely eliminated and the detection accuracy show an obvious improvement. In addition, the position accuracy also outperforms the DSOD (Figure 8(c2)).

### 3.4. Comparison with Other State-of-the-art Methods

To illustrate the effectiveness of our proposed algorithm framework, we carried out a series of comparative experiments with other state-of-the-art methods. For ship detection from remote sensing image, we combined the scene extraction and the classical detection algorithm with different combination strategies. The scene was generated using the sliding window (SW) and the $L^{2}$CSN. The five combinations are as follows: SW+SSD, SW+DSOD, $L^{2}$CSN+SSD, $L^{2}$CSN+DSOD and the proposed method. In the comparative experiment, we set the network parameters of all the methods in accordance with our proposed method, including the training set and validation set.

As shown in Figure 9, we used the SW+DSOD and our proposed method to detect a typical remote sensing image. According to the experimental results, the proposed $L^{2}$CSN effectively inhibits the false alarms in the non-ship area, and the proposed VA-DSOD enhances the detection accuracy in the local candidate scenes.

We further evaluated the detection performance of different methods with 100 test images, calculating the *precision rate* and *recall rate* of each methods. The *precision rate* ($P_{d}$) and the *recall rate* ($R_{d})$ are calculated by the following:(8)$$P_{d}=\;\frac{TP^{d}}{TP^{d}+FP^{d}}=\frac{\sum_{i}^{N}TP_{i}^{d}}{\sum_{i}^{N}(TP_{i}^{d}+FP_{i}^{d})}$$
(9)$$R_{d}=\;\frac{TP^{d}}{TP^{d}+FN^{d}}=\frac{\sum_{i}^{N}TP_{i}^{d}}{\sum_{i}^{N}(TP_{i}^{d}+FN_{i}^{d})}$$
where $TP^{d}$ represents the number of correctly detected ships,$\;FP^{d}$ represents the number of falsely detected ships, $FN^{d}$ represents the number of missed ships, and N indicates the number of validation image.

We can obtained different $P_{d}$ and $R_{d}$ by changing the score threshold of the detection results, which are represented in the form of $P_{d}-R_{d}$ curves, as shown in Figure 10. From the $P_{d}-R_{d}$ curves, the detection performance of $L^{2}$CSN +SSD, $L^{2}$CSN+DSOD and our proposed method significantly outperformed SW+SSD and SW+DSOD. Compared with the methods, without the prejudging of local candidate scenes, the $L^{2}$CSN enhanced the accuracy of the detection framework by suppressing false alarms in non-ship areas. $L^{2}$CSN+DSOD can achieve better detection performance than $L^{2}$CSN+SSD, and we can clearly see that DSOD is superior to SSD, even without the base network of pre-training. It can be seen that our proposed method outperforms other methods, which can be attributed to the fact that they have stronger of features representation abilities. 

Through quantitative results with different detection methods, as shown in Table 1, we can see that the $R_{d}$ and $P_{d}\;$ of the proposed method all show an obvious improvement, and the value of average precision (AP) increased by 7.53%. Meanwhile, we evaluate the computational time costs of the different detection methods, and all the methods are run on a GTX Titan X GPU (12 GB on board memory). We tested a remote sensing image of typical scene (with 6370 × 11860 pixels) using different methods. Our proposed method did not increase more time in comparison with $L^{2}$CSN+DSOD, and the detection performance and timeliness of the proposed method is also higher than other methods.

## 4. Conclusions

In this paper, we propose a ship detection method from optical remote sensing images, based on visual attention enhanced network. Frist, we develop the $L^{2}$CSN to extract the local candidate scenes, which largely reduces the false alarms in non-ship areas and improves the detection efficiency for remote sensing images. Then, for the extracted local candidate scenes, we structure a ship detection method, based on VA-DSOD, which extracts semantic features and visual features, based on VA-DSOD to improve the detection performance. Additionally, the rotated bounding box regression is employed with the VA-DSOD to enhance the positioning accuracy. 

The proposed method has been compared with the other state-of-the-art methods, using a large number of dataset with ship scenes. The qualitative experiment results indicate that the detection performance of the proposed methods are also better than other methods, especially in the detection of docked ships and side-by-side ships. The rotated bounding box regression based on VA-DSOD also enhances positioning accuracy. Through quantitative experiment results, we can see that recall and precision of the proposed method all show an obvious improvement, and the value of AP increases by 7.53%. Meanwhile, our proposed method does not increase more time in comparison with other methods. In the future, we plan to use a visual attention features enhance network in the spatial dimension, and to further implement end-to-end of networks.

## Figures and Tables

**Figure 1 sensors-19-02271-f001:**
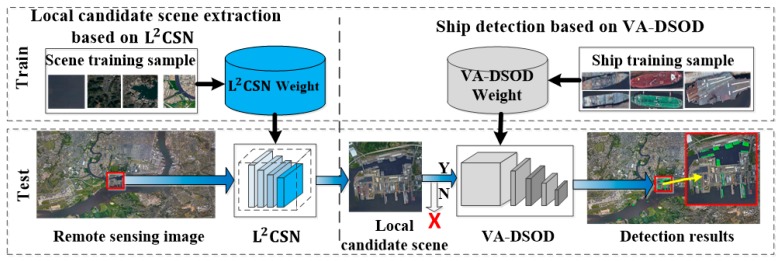
The framework of the proposed method.

**Figure 2 sensors-19-02271-f002:**
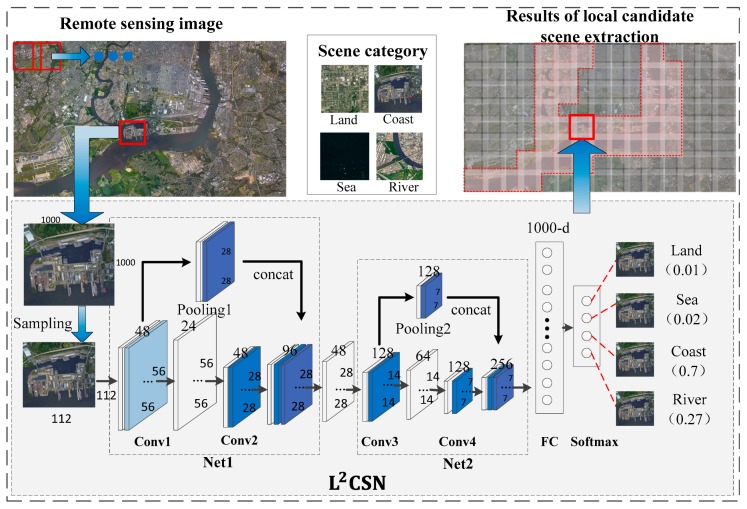
The $L^{2}$CSN architecture.

**Figure 3 sensors-19-02271-f003:**
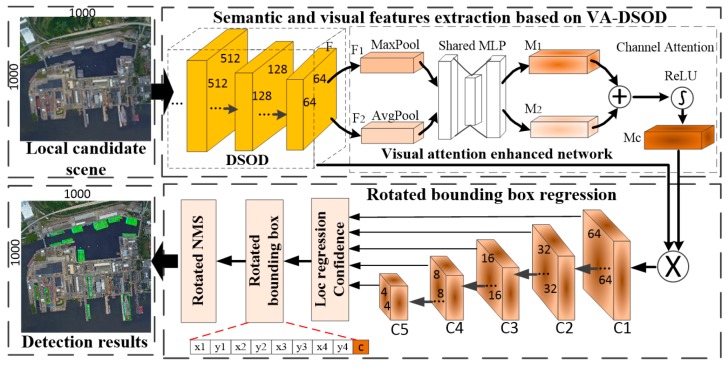
The flow chart of ship detection based on VA-DSOD.

**Figure 4 sensors-19-02271-f004:**
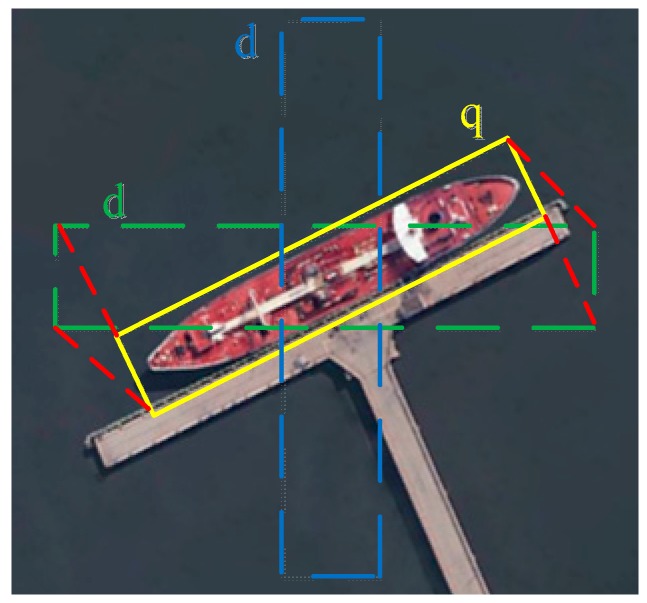
Rotated bounding box regression.

**Figure 5 sensors-19-02271-f005:**
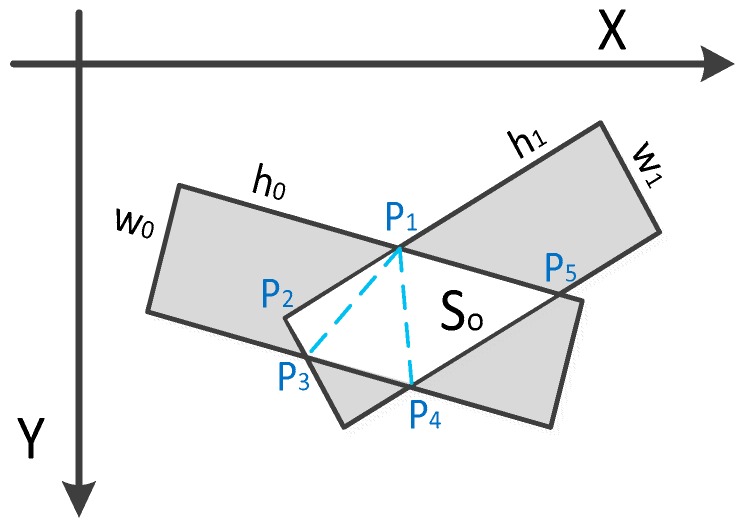
Rotated NMS.

**Figure 6 sensors-19-02271-f006:**
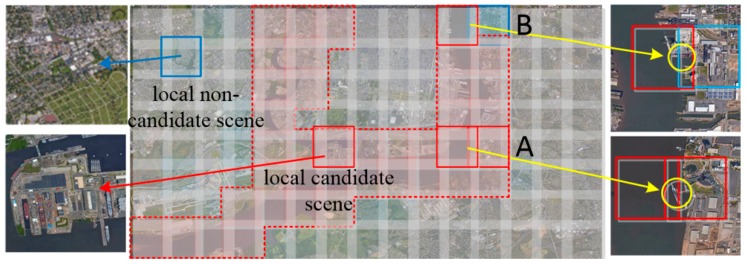
Extraction results for local candidate scenes using the $L^{2}$CSN.

**Figure 7 sensors-19-02271-f007:**
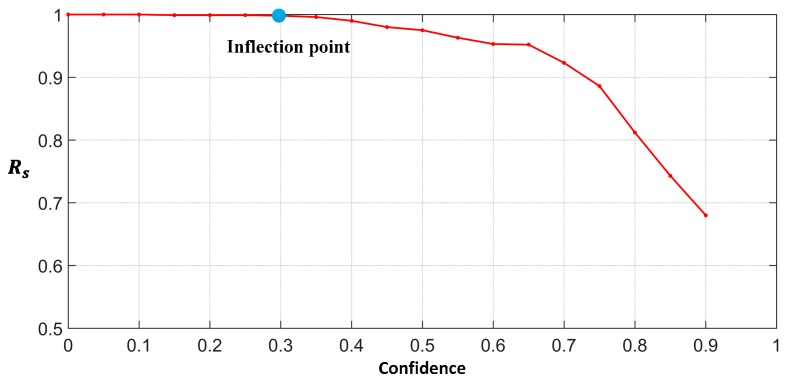
The $R_{s}$ curves of the local candidate scenes.

**Figure 8 sensors-19-02271-f008:**
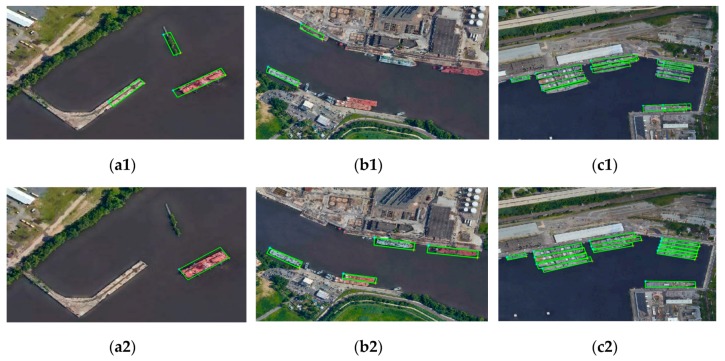
Ship detection results in typical scenes. (**a1**,**b1**,**c1**) of DSOD, (**a2**,**b2**,**c2**) of VA-DSOD.

**Figure 9 sensors-19-02271-f009:**
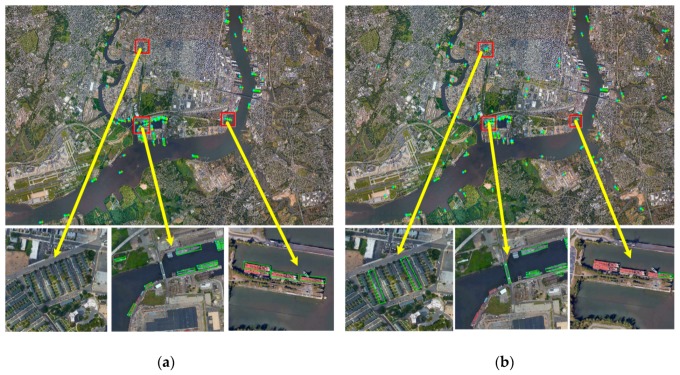
The comparisons of our detection results with different combination strategies. (**a**) Our proposed method, (**b**) SW+DSOD.

**Figure 10 sensors-19-02271-f010:**
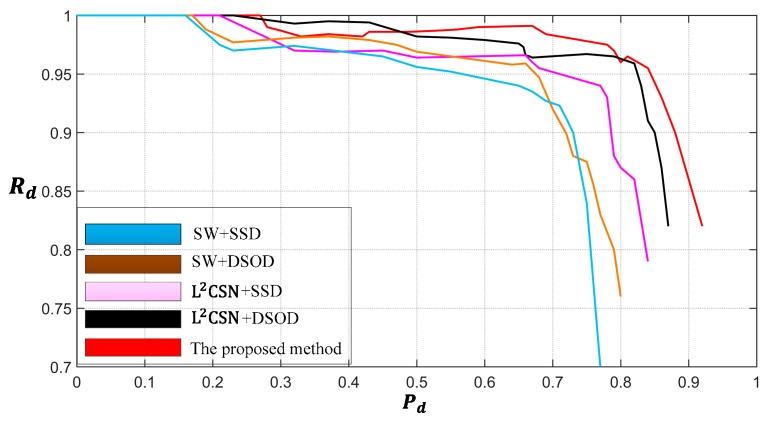
The $P_{d}-R_{d\;}$ curves of the different detection methods.

**Table 1 sensors-19-02271-t001:** Quantitative results with different detection methods (IoU = 0.4, Score = 0.5).

Detection Methods	SW+SSD	SW+DSOD	$L^{2}$CSN+SSD	$L^{2}$CSN+DSOD	Proposed Method
**Time costs (s)**	7.621	10.308	3.367	4.433	3.605
$R_{d}$	0.755	0.752	0.792	0.837	0.843
$P_{d}\;$	0.848	0.875	0.891	0.943	0.954
**AP(%)**	79.05	82.33	84.47	87.72	89.86

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
