# Peer review of "Ship Detection for Optical Remote Sensing Images Based on Visual Attention Enhanced Network"

_sensors, 2019, doi:10.3390/s19102271_

Round 1
Reviewer 1 Report
This paper proposes a ship detection method for optical remote sensing image based on visual attention enhanced network. Through quantitative and qualitative experiments, the proposed method is validated in terms of the timeliness and detection performance in complex scenes. Some suggestions are as below: 1. As aruged by authors, the ship detection always involves a small data size issue. Thus, besides the proposed method, could some shallow feature extraction methods (or other specified deep methods) be used as a baseline? e.g. Multiple Marginal Fisher Analysis; Automatic Subspace Learning via Principal Coefficients Embedding; Locality Adaptive Discriminant Analysis for Spectral-Spatial Classification of Hyperspectral Images; Embedding Structured Contour and Location Prior in Siamesed Fully Convolutional Networks for Road Detection. It is highly expected to see the comparsion with these method, at least some discussions are preferable. 2. How to determine the parameters of the proposed method? It would be better to investigate the inflence of parameters, as well as conducting some abaltion studies on the compoenents.Author Response
Dear reviewer:
Thank you for your comments and suggestions, sensors-495702 Type of manuscript: Article Title: Ship Detection from Optical Remote Sensing Images based on Visual Attention Enhanced Network. We have made the revisions according to the helpful comments and suggestions and uploaded our response in the form of PDF.
Thank you very much.
Best regards
Fukun Bi , Jinyuan Hou , Liang Chen , Zhihua Yang and Yanping Wang

Reviewer 2 Report
Dear authors,
Thank you for a novel contribution. Methodologically, this manuscript has scientific merit. However, it lacks the strong literature review in the introduction and contrast with previous works. For example, contributions in convolutional neural networks in particular using optical remote sensing products would warrant citations and contrast.
Few recommendations below:
Ghorbanzadeh, O., Blaschke, T., Gholamnia, K., Meena, S. R., Tiede, D., & Aryal, J. (2019). Evaluation of different machine learning methods and deep-learning convolutional neural networks for landslide detection. Remote Sensing, 11(2), 196.
Dutta, R., Aryal, J., Das, A. and Kirkpatrick, J.B., 2013. Deep cognitive imaging systems enable estimation of continental-scale fire incidence from climate data. Scientific reports, 3, p.3188.
Peng Ding, Ye Zhang, Wei-Jian Deng, Ping Jia, Arjan Kuijper. (2018). A light and faster regional convolutional neural network for object detection in optical remote sensing images, ISPRS Journal of Photogrammetry and Remote Sensing,141, 208-218.
The presentation needs an improvement in particular the figure configuration and equations.
I will be happy to review the next iteration.
Thanks.
Author Response
Dear reviewer:
Thank you for your comments and suggestions, sensors-495702 Type of manuscript: Article Title: Ship Detection from Optical Remote Sensing Images based on Visual Attention Enhanced Network. We have made the revisions according to the helpful comments and suggestions and uploaded our response in the form of PDF.
Thank you very much.
Best regards
Fukun Bi , Jinyuan Hou , Liang Chen , Zhihua Yang and Yanping Wang

Reviewer 3 Report
This paper presents a local candidate scene network to extract the local candidate scenes with ships, and then proposes a ship detection method based on the visual attention DSOD (VA-DSOD). Although the quite good work presented, the article is poorly written in terms of grammar and sentence structure. I suggest sending it to an English editor. Comments are listed below:
1. Modify the title to: Ship Detection from Optical Remote Sensing Images based on Visual Attention Enhanced Network.
2. Line 13: what does wide field of image mean?
3. Line 15 and Line 16: for > from
4. In the abstract, give some numbers on the results.
5. Line 30: in > from
6. Line 37: construct > constructed
7. Line 38: remove using; is > was
8. The footer/header is written as “Remote Sens. 2019, 11, x FOR PEER REVIEW …” This article was submitted to Sensors I believe.
9. Line 52: “Most of the above…” repeated sentence and very generic. Remove it.
10. Line 55: “and all of the above …” please be specific and list these factors.
11. Line 72: ship > a ship, for > from
12. Line 75: is extracts?
13. Line 108: 5000*5000 is it pixels? Please indicate in the text.
14. Line 109: 1k*1k? Do you mean 1000*1000, please be consistent.
15. Line 169 and Line 172: from where did you get those values? these scales and ratios require a reference.
16. Line 204 and Line220: LLSCN?
17. Line 209,210: Google images? What is the source of Google images and their resolutions?
18. Line 209: 6410 from Google images and GeoFen-2. How many of each?
19. Line 212: manually divided into four classes, this is questionable. How did you divide all these images manually? And why manually? Have you tried classification technique?
20. Line 223: 6370*11860, what is this dimension?
21. Line 239: what do you mean by confidence?
22. What is the value of Rs of the validation data?
23. Figure 10, it is better to draw a bar chart or convert it to table.
24. Support the conclusions by results.
Author Response

(The authors gave the same response as above.)

Round 2
Reviewer 1 Report
most concerns of mine have been addressed.Author Response
Dear reviewer:
Thank you for your comments and suggestions again, sensors-495702 Type of manuscript: Article Title: Ship Detection from Optical Remote Sensing Images based on Visual Attention Enhanced Network. Your valuable comments and suggestions are very helpful to improve our article. We have made the revisions according to the helpful comments and suggestions, the detailed responses are given item-by-item below.
Thank you very much.
Best regards
Fukun Bi , Jinyuan Hou , Liang Chen , Zhihua Yang and Yanping Wang
Response to Reviewer Comments
Point 1: Most concerns of mine have been addressed.
Response 1: Thank you for your careful reading and comments. According to your suggestions, we have made reasonable modifications to our article, which make it more readable and logical. We have also made more detailed and minor changes for the charts and gramma, etc.
Thank you very much.
Best regards
Fukun Bi , Jinyuan Hou , Liang Chen , Zhihua Yang and Yanping Wan
Reviewer 2 Report
Dear authors,
Thank you for addressing all the comments, appreciated.
I advise to rewrite the final sentence of the abstract for further clarity on the improvement of the proposed method against the previous method. You mentioned the increase in performance, would be great to write the original percentage and the change in percentage value.
Others are OK.
Regards
Author Response
Dear reviewer:
Thank you for your comments and suggestions again, sensors-495702 Type of manuscript: Article Title: Ship Detection from Optical Remote Sensing Images based on Visual Attention Enhanced Network. Your valuable comments and suggestions are very helpful to improve our article. We have made the revisions according to the helpful comments and suggestions, the detailed responses are given item-by-item below.
Thank you very much.
Best regards
Fukun Bi , Jinyuan Hou , Liang Chen , Zhihua Yang and Yanping Wang
Response to Reviewer Comments
Point 1: Advise to rewrite the final sentence of the abstract for further clarity on the improvement of the proposed method against the previous method. You mentioned the increase in performance, would be great to write the original percentage and the change in percentage value.
Response 1: Thank you for your careful reading and comments. The final sentence of the abstract lack the detailed description. According to your suggestions, we have added a description in line 23.
We regard the state-of-the-art method [sliding window DSOD (SW+DSOD)] as a baseline, which achieves the average precision (AP) of 82.33%. The AP of the proposed method increases by 7.53%. The detection and location performance of our proposed method outperforms the baseline in complex remote sensing scenes.
Thank you very much.
Best regards
Fukun Bi , Jinyuan Hou , Liang Chen , Zhihua Yang and Yanping Wang
Reviewer 3 Report
On replying on Point 17, you did not mention the source of images (satellite images e.g., worldview or optical images). If you cannot reach the exact source, please mention in the text the software you used to download these images.
On replying on Point 19, if I understood it correctly that the company provided the manual classification, please mention that in the text.
Line 281-284: I cannot understand what you really want to tell. Please try to write in a better way.
Author Response
Dear reviewer:
Thank you for your comments and suggestions again, sensors-495702 Type of manuscript: Article Title: Ship Detection from Optical Remote Sensing Images based on Visual Attention Enhanced Network. Your valuable comments and suggestions are very helpful to improve our article. We have made the revisions according to the helpful comments and suggestions, the detailed responses are given item-by-item in PDF.
Thank you very much.
Best regards
Fukun Bi , Jinyuan Hou , Liang Chen , Zhihua Yang and Yanping Wang
